# Salt-Induced Changes in Cytosolic pH and Photosynthesis in Tobacco and Potato Leaves

**DOI:** 10.3390/ijms24010491

**Published:** 2022-12-28

**Authors:** Anna Pecherina, Marina Grinberg, Maria Ageyeva, Daria Zanegina, Elena Akinchits, Anna Brilkina, Vladimir Vodeneev

**Affiliations:** 1Department of Biophysics, National Research Lobachevsky State University of Nizhny Novgorod, 23 Gagarin Avenue, 603950 Nizhny Novgorod, Russia; 2Department of Biochemistry and Biotechnology, National Research Lobachevsky State University of Nizhny Novgorod, 23 Gagarin Avenue, 603950 Nizhny Novgorod, Russia

**Keywords:** salinity, photosynthesis, cytosolic pH, sodium, fluorescence imaging, transpiration, tobacco, potato

## Abstract

Salinity is one of the most common factors limiting the productivity of crops. The damaging effect of salt stress on many vital plant processes is mediated, on the one hand, by the osmotic stress caused by large concentrations of Na^+^ and Cl^−^ outside the root and, on the other hand, by the toxic effect of these ions loaded in the cell. In our work, the influence of salinity on the changes in photosynthesis, transpiration, water content and cytosolic pH in the leaves of two important crops of the *Solanaceae* family—tobacco and potato—was investigated. Salinity caused a decrease in photosynthesis activity, which manifested as a decrease in the quantum yield of photosystem II and an increase in non-photochemical quenching. Along with photosynthesis limitation, there was a slight reduction in the relative water content in the leaves and a decrease in transpiration, determined by the crop water stress index. Furthermore, a decrease in cytosolic pH was detected in tobacco and potato plants transformed by the gene of pH-sensitive protein Pt-GFP. The potential mechanisms of the salinity influence on the activity of photosynthesis were analyzed with the comparison of the parameters’ dynamics, as well as the salt content in the leaves.

## 1. Introduction

Salinity is one of the most common abiotic stressors limiting the agriculture industry [1], mainly due to the prevalence of saline lands and the disruption of irrigation [2,3] and due to most crops being glycophytes [4]. Salinity causes a decline in plant growth and development and reduces plant productivity. The expression of plant response depends on the concentration of salinity agents, types of crops and the stage of plant development [5,6,7]. Currently, the selection of salt-resistant crops is being actively performed, and the success of this selection depends on the study of the fundamental mechanisms of stress resistance in plants, which requires in-depth study [4,8].

Salt stress is a complex effect that includes osmotic and ionic components. The distinctive feature of the osmotic component is the reduction in the potential of water caused by Na^+^ and Cl^−^, which is the first reaction of plants and spreads rapidly from roots to shoots [9,10]. The next phase is the increase in the absorption of Na^+^ and the disruption in the ion balance caused by ion stress [9,11,12]. The concentration of sodium in the cytosol does not normally exceed 10 mM [12,13]. High extracellular concentrations of Na^+^, due to the high gradient of the electrochemical potential of Na^+^, lead to its entry into the cytosol of root cells, mainly through nonselective cation channels (NSCCs). A decrease in intracellular concentration occurs due to the pumping of Na^+^ from the cytosol into the xylem, which is performed by Na^+^/H^+^-antiporter SOS1, or into the vacuole, which is performed by Na^+^/H^+^-antiporter NHX [13,14,15,16]. Sodium is loaded from the xylem into the leaf cells via NSCCs and HKT transporters; SOS1 and NHX excrete sodium from the cytosol into the phloem [4,9,17].

Sequentially developing osmotic and ionic components cause the observed multiple effects of salinity, including a decrease in water and nutrient absorption, protein synthesis, signal transmission, etc. However, the identification of an association between the response of the plant and a certain component of salt stress has yet to be clarified.

Much research has been devoted to studying the effects of salt on photosynthesis, because photosynthesis is the most important physiological process that determines plant productivity. On the one hand, the effect of salinity on photosynthesis is associated with the influence of osmotic stress on the stomatal apparatus (stomatal limitations) [18] and, on the other hand, ionic toxicity leads to non-stomatal limitations [19,20].

Salinity causes disruption of the water flow into the leaves and leads to the synthesis of abscisic acid. These effects induce the closure of stomata and a decrease in CO_2_ uptake [21,22,23]. The progressive accumulation of salt and the violation of the intracellular homeostasis of potassium and calcium [8,10,22] cause non-stomatal limitations in leaves: the further inhibition of CO_2_ assimilation, the suppression of the activity of Calvin–Benson cycle enzymes, the disruption of chlorophyll biosynthesis and the decrease in the efficiency of the functioning and structural integrity of the photosynthetic apparatus and thylakoid membranes [23,24,25,26,27,28]. Presumably, one of the results and mechanisms of ion toxicity is increased reactive oxygen species (ROS) generation in photosystem I (PSI) and photosystem II (PSII) [29]. Salt stress evokes damage to the oxygen evolving complex (OEC) and blocks electron transfer from water to PSII, causing the inhibition of the quantum yield of PSII [30,31,32]. Along with ROS, other cellular pathways and mechanisms are involved in the regulation of activity of photosynthetic processes and one of these regulators could be cytosolic pH [23,33,34].

Salinity leads to a change in cytosolic pH, which has a different character depending on the duration of salinity. In the first stages (several minutes–days) of salinity, the cytosolic pH decreases [35,36,37]. Primarily, this fact can be explained by the inhibition of the proton pumps possibly involving FERONIA kinase [38,39]. In addition, a decrease in cytosolic pH may be associated, on the one hand, with the influx of Cl^−^ and H^+^ to the cytosol via Cl^−^/H^+^-cotransporters CCC [10,40,41] and, on the other hand, with the desire of the cell to remove Na^+^ from the cytosol using Na^+^/H^+^-exchangers [42,43]. Further, the cytosol alkalizes [44,45], which is associated with an increase in the activity or expression of proton pumps [46,47].

It is worth noting that most studies of the effect of salinity on pH have been conducted on plant roots or protoplasts, while very few studies of the pH level in the leaves have been carried out. Indirectly, the acidification of the leaf cytosol can be proven by apoplast alkalinization as described by [41,48,49]. In addition, other research studies practically do not cover the issue of the relationship between intracellular pH and photosynthesis under salinity conditions. In addition, the contribution of leaf location at different heights in shoots (strata) to the response of photosynthesis to salinity has been poorly studied. In this work, we evaluated the temporal dynamics of photosynthesis activity, transpiration, cytosolic pH, and water and sodium content in response to salinity in the leaves of different stratum in two important crops—tobacco and potato. Based on a comparison of the dynamics of these processes, the osmotic or ionic component of salt stress was determined to be responsible for changing the level of cytosolic pH and photosynthesis activity.

## 2. Results

### 2.1. Effect of Salinity on Photosynthesis

The effect of soil salinity on the photosynthetic processes of tobacco and potato plants was studied on the leaves of three different types (strata): 1—the oldest leaf located in shoot closer to the soil; 2—mature leaf; 3—the youngest of the fully formed leaves (Figure 1A and Figure 2A). Some parameters of photosynthesis were analyzed: PSII fluorescence in the dark-adapted state (F_v_/F_m_), quantum yield of the photosystem II (Ф_PSII_), non-photochemical quenching (NPQ) and chlorophyll index (CHL-Ind). The analysis of these parameters revealed that photosynthesis was a little more intense in young leaves than in older ones under salinity (Figure 1B and Figure 2B) and non-treatment conditions (Appendix A, Appendix A). Additionally, a long-term measure of these parameters could be accompanied with a slight drift in water-treated control plants; in this regard, the analysis of the effect of soil salinity on photosynthetic processes was evaluated by the difference of each parameter in salt-treated and control plants (Figure 1C and Figure 2C).

Parameters of the photosystem II activity changed in tobacco leaves under salinity. F_v_/F_m_, which indicates the intact PSII structure, decreased within a little range (0.01–0.03) (Figure 1C, Appendix A). Structural changes have occurred in the second and third leaves earlier (after 24 h) than in the first leaf (48 h).

Ф_PSII_ decreased faster and stronger for the second and third tobacco leaves than for the first leaf. Significant differences in comparison with the control were noted for these leaves already within 10 h after the start of salt stress, and for an older first leaf—after 24 h (Figure 1C). Salinity has enhanced NPQ: significant changes occurred most quickly in the second leaves (after 10 h) and in other leaves—after 24 h (Figure 1C). Salt-induced changes in activity of the PSII were similar in direction in tobacco leaves of different stratum; however, for younger leaves (3 and 2) they occurred earlier than for the older leaf. CHL-Ind has not changed in comparison with the non-treated plants (Appendix A).

Salinity has also led to a change in the parameters of the PSII activity in potato plants (Figure 2). A significant decrease in the F_v_/F_m_ for the first leaf was detected at 34 h after the start of salt stress; in the second and third leaves changes had occurred later. The amplitude of the change in potato leaves was greater than in tobacco and the value had reached 0.20 units (Figure 2C, Appendix A).

Ф_PSII_ most significantly decreased for leaves 2 and 3, significant differences with control plants were identified at 10 h of salt stress. On the contrary, NPQ had a faster increase (started at 5 h of salt stress) in the older first leaf, although stronger increase in the NPQ was detected for the third leaf at one day and more after the start of salinity. The chlorophyll index did not change in comparison with the control (Appendix A).

Similarly, we have identified unidirectional changes in the parameters of the PSII activity in tobacco and potato leaves. For both plants, stronger changes highlighting the negative effect of salt have occurred in younger leaves that were positioned higher than old leaf. A large response amplitude of photosynthesis was observed for potato plants in comparison with tobacco, especially for F_v_/F_m_, indicating a greater sensitivity of potato to salinity.

### 2.2. Stomatal Conductance and Water Content in Leaves after Start of the Salt Stress

The effect of salinity on transpiration and water content in leaves was assessed by crop water stress index (CWSI) and relative water content (RWC) index, respectively. The CWSI was calculated by measuring the temperature of the leaf surface. The surface temperature of tobacco leaves had increased in comparison with control plants during salinity. This fact indicated an increase in the number of closed stomata (Figure 3A). Transpiration had significantly decreased in all leaves at 10 h of salt stress and remained during the experiment for 2d and 3d leaves (Figure 3B).

RWC in tobacco leaves had significantly decreased at 10 h of salinity treatment on the first leaf, and 24 h later on the second and third leaves (Figure 3C); however, the trend of this reduction was observed already after 5 h of salt stress. This fact is also consistent with our visual observations: at the end of the first day, salinity treatment and leaves (especially 1 and 2 leaves) had lost their turgor. The leaves regained their turgor by the end of the second experimental day.

CWSI had more dramatically reduced in potato leaves during salinity. A significant decrease in transpiration in all leaves was observed at 3 h after salt application, and this change persisted for 48 h of observation. This change was stronger for the younger leaf (Figure 4B), and was consistent with a smaller number of open stomata. We also noted that CWSI of potato leaves was the first one with significant changes after the start of salt treatment.

The relative water content of the first potato leaf had significantly decreased at 24 h of salinity and for the second leaf—at 48 h (Figure 4C). RWC did not significantly change in the third leaf during salt treatment (Figure 4B).

### 2.3. Change of Leaves’ pH during Salinity

The cytosolic pH levels of tobacco and potato leaves were determined using a ratiometric genetically encoded sensor Pt-GFP constantly expressed in these plants. For determination, the ratio of the fluorescence intensity from the anionic (excitation with a maximum at 475 nm) to the neutral form (excitation with a maximum at 405 nm) of the Pt-GFP was calculated (Figure 5A and 6A). The pH values in the cytosol of leaf cells were evaluated by calibration dependence (Appendix A). Leaves of different stratum had similar initial levels of cytosolic pH (Appendix A).

Salinity had caused cytosol acidification of the tobacco leaves, which lasted during the entire period of salt treatment. The earliest significant changes occurred for the oldest leaf at 9 h of salinity (Figure 5B). In the second and third leaves, cytosolic pH significantly decreased at 48 h after salt treatment. The amplitude of pH decrease was 0.5 pH units at the end of the experiment.

Salinity had also caused acidification of the leaf cytosol in potato plants (Figure 6B). 

Younger potato leaves turned out to be more salt-sensitive than other leaves; in younger leaves, significant differences of pH were observed 9 h after the start of salinity treatment. At the same time, a maximal reduction in cytosolic pH (more than 1 pH unit) was detected for the youngest leaf at two days after the salt addition.

### 2.4. Sodium Content in Leaves

The treatment of soil by the salt solution caused the accumulation of sodium in the leaves of potato and tobacco. These crops differed in the rate of sodium influx into the leaves. A significant increase in the sodium content in tobacco leaves occurred after 24 h of salt stress and increased depending on the leaf type. The higher the leaf was located in the shoot, the higher the storage of sodium was over time (Figure 7A).

Accumulation of sodium was simultaneous in all potato leaves at 10 h after the start of salt treatment. Sodium was loading into the second leaf more intensively; this fact is consistent with the parameters of water exchange for this leaf (Figure 4 and Figure 7B). Dynamics of chloride accumulation have a similar rate to the sodium loading into the leaves (Appendix A).

## 3. Discussion

Tobacco and potato, being glycophytes, are among the model examples for studying the mechanisms of salinity effect on plants [50,51]. These plants are used in many research studies which describe salt stress, and this fact allows us to compare data from previous research with our results. Most often, it is noted that salinity negatively affects the fresh and dry weight of plants, leaf area, relative water content in leaves, photosynthesis and transpiration activity, and it also changes the activity of proton pumps [12,26,52,53,54].

In our work, photosynthesis activity decreased in response to salinity: Ф_PSII_ reduced (the amplitude was up to a maximum of 0.2–0.3 units) and NPQ increased (~0.5 units). The amplitude of changes in these parameters in tobacco and potato is consistent with the results of other studies with treatment using similar salt concentrations [26,54,55]. A significant decrease in Ф_PSII_, which indicates the efficiency of using of the light absorption energy by chlorophyll [56], occurred at 10 h of salt stress in tobacco (Figure 1C) and potato plants (Figure 2C). NPQ, characterizing the thermal energy losses of absorbed light [56], significantly increased at 5 and 10 h of the salt treatment in potato and tobacco plants, respectively. What was defined in the work of the temporal dynamics of the reduction in the photosynthesis activity, was not contrary to previous research studies [21,26,54,55,57].

The main ways in which salinity influences the activity of photosynthesis were determined in this study and are owed great attention from researchers, as photosynthesis is a process which plays a key role in plant growth. The activity of photosynthesis is affected by the osmotic and ionic components of salt stress [22]. One of the main methods of the influence can be a decrease in stomatal conductivity leading to a decrease in CO_2_ entrance [21,22,23]. A decrease in the CO_2_ input to cells leads to a reduction in the activity of Calvin–Benson cycle enzymes and, accordingly, a decline in the consumption of ATP and NADPH [58]. These changes cause suppression of the light-driven reactions of photosynthesis [59]. These consequences of transpiration decrease cause a restriction of the electron flux from PSII to PSI, an obstacle to cyclic electron transport around PSI and cytochrome b6f that cause violence of the photoprotection, carotenoid damage and increase of non-photochemical quenching [21,23,24,25,26,27,28]. The ionic component of salinity also has a negative effect on photosynthesis due to the toxicity of Na^+^ and Cl^−^ and the disruption of intracellular homeostasis of K^+^ and Ca^2+^ caused by them [22]. Sodium inhibits protein D1, protein 1 of OEC, ATP synthase subunits, Rubisco and fructose-1,6-bisphosphatase [22,60,61]. In normal conditions, Cl^−^ participates in the regulation of many processes including photosynthesis and evidence of its toxicity is poorly known; however, Cl^−^ causes degradation of chlorophyll and reduces the activity of photosynthesis [62,63,64]. The lack of potassium contributes to the weakening of CO_2_ fixation in the Calvin–Benson cycle and an increase in ROS production [10,47,65,66]. Inhibition of the electron transport chain, caused by a decrease in the rate of CO_2_ utilization, leads to the pseudo-cyclic electron transport, which contributes to excessive accumulation of ROS and further damage to the photosynthetic apparatus [21,24,28].

In our study, a decrease in the value of F_v_/F_m_, corresponding to the proportion of intact (actively functioning) photosystems [56], occurred later than the change in parameters of photosynthesis activity (Figure 8). The amplitude of changes and time interval (no earlier than a one day) between the start of salt treatment and the start of F_v_/F_m_ changes correlated with the previous works for potato [54,67,68] and tobacco [21,26,57,69].

For identification of the most probable causes of photosynthesis inhibition induced by salinity, we compared the temporal dynamics of parameters which characterize the ionic and osmotic components of salt stress, along with photosynthetic activity (Figure 8).

The analysis of temporal dynamics indicates that the ionic component of salinity (the accumulation of Na^+^ in the leaves) apparently is not the reason for the decrease in photosynthetic activity at the initial stages of salt stress, since Na^+^ accumulation occurred later than when photosynthetic activity started to decrease. At the same time, the F_v_/F_m_ reduction coincided (for tobacco) or occurred later (for potato) than when sodium started accumulating; therefore, sodium could have a damaging effect on the photosynthetic apparatus (Figure 8). The potassium content slightly decreased after only 2 days in tobacco (Appendix A), and this fact corresponded to the results of the other study with a similar temporal dynamic [68]. Consequently, the concentration of potassium had minimal effects on photosynthesis in our research.

The osmotic component of salinity can be assumed as the most possible cause of a decrease in the photosynthesis activity at the initial stages (Figure 8), due to the most rapid change during salinity in our study. The water homeostasis of the leaves, changing by the osmotic component, was evaluated by the relative water content and the change in transpiration, which was estimated in our work using the CWSI by measure of leaf temperature [70,71]. The identified dynamics of salt-induced changes in transpiration corresponded to results described in other studies on potato [17,72], tomato [26], maize [68] and tobacco [63].

Early changes in photosynthetic activity could also be the result of intracellular pH decrease because the time of pH change was similar to the change in stomatal conductivity and RWC. Acidification of the cytosol can inhibit carbonic anhydrase affecting the CO_2_ availability [73]. Possible acidification of cytosolic pH may affect the transport of NAD^+^ by carrier protein NDT1 into chloroplasts [74]. NADPH^+^ is required for PSI and ATP synthesis, and reduction in its content leads to inhibition of the light-driven and light-independent photosynthesis processes [75]. The lack of NAD^+^ in chloroplasts can also cause the production of glycolaldehyde, an inhibitor of many enzymes of the Calvin–Benson cycle [59,76]. To sum up, the cytosol acidification, which was observed in the early stages, may be a factor inducing photosynthesis decrease.

The role of pH in the regulation of a wide range of processes in the cell and energization of membrane transport is important, which is supported by a large number of research studies have been devoted to the study of pH dynamics under various stresses including salt stress [35,37,49,77,78,79,80]. Previous studies have mainly focused on the research of cytosolic [78,81,82], vacuolar [83] and apoplastic pH and the proton fluxes [37,84,85,86] in root cells. Studies of the salt-induced pH shifts in leaves were performed only for apoplast [35,38,49,79,87,88], and data from these studies cannot be compared to our results. The effect of salinity on the leaf cytosolic pH was studied mainly at the cellular level or at isolated protoplasts; the salt-induced acidification of the cytosol occurred in the range from 0.4 to 1.3 pH units, and it coincides with our values [36,37,88].

In this case, we can assume several reasons for the change in cytosolic pH in leaves caused by adding salt to the roots. A possible reason for the early decrease in cytosolic pH is the inactivation of proton pumps by Ca^2+^-dependent protein kinases or by FERONIA [10,89,90] due to the propagation of a remote stress signal from the roots to the shoots—mutually interplayed waves of calcium and ROS [9,15,47,91]. An increase in the concentration of cytosolic calcium in the roots occurs through depolarization-activated non-selective cation channels (DA-NSCCs) by the osmotic component of salinity [9,15,72,92,93], or due to unknown calcium channels that are activated by sodium-binding glycosyl inositol phosphoryl ceramides (GIPCs) on the plasmalemma [94,95].

A long-term pH decrease in leaves may also be due to the ionic component of salt stress—sodium input to the leaf (FERONIA can physically interact with sodium in pectin and inhibit H^+^-ATPase [39,95])—or excretion from the leaf cytosol (Na^+^/H^+^-exchangers load protons into cytosol [5]). Additionally, some researchers highlight the influx of chlorine with protons along the Cl^−^/H^+^-symporters CCC as a possible mechanism for reducing cytosolic pH [41,48]. The ionic component most likely did not play any role at the early stages of pH decreasing; the accumulation of sodium ions in the leaf occurred later, as well as the intake of Cl^−^ by CCC symporters, since the rate of input of sodium and chlorine from the roots is the same [53,96] and this fact was confirmed in our study (Appendix A). Further cytosolic pH changes have occurred due to the activation (halophytes) or increased expression (glycophytes) of proton pumps [97]. An increase in proton efflux results in cytosol alkalinization, which is necessary to feed the proton-motive force of Na^+^/H^+^-antiporters to remove Na^+^ from the cytosol [45].

Comparison of salt-induced changes in the indicators/parameters of the leaves of different types showed that stronger changes occurred in the leaves located higher on the shoot (Figure 9). The leaves located lower on the shoot were older and apparently this fact negatively affects the initial activity of processes, including photosynthesis [98,99,100]. In addition to high photosynthetic activity, young leaves are also characterized by a higher rate of vascular transport [101] causing stronger response to salt stress.

Thus, in our work we demonstrated the dynamics of several indicators in situ, including photosynthesis activity, intracellular pH and transpiration. Early changes in these parameters are caused by osmotic components and, possibly, long-distant signals from roots. A decrease in cytosolic pH may be one of the factors resulting in the reduction of photosynthesis. The accumulation of ions only prolongs these effects and has a toxic destructive effect on the photosynthetic apparatus.

## 4. Materials and Methods

### 4.1. Plant Material and Salt Treatment

Plants of *Solanum tuberosum* (variety Nevsky, transgenic line NKM01) and *Nicotiana tabacum* (variety Samsun, transgenic line 177/3) constantly expressing fluorescent pH-sensitive sensor Pt-GFP [82] were used in the experiments. The procedure of agrobacterial transformation of potato was described in previous study [102]. The transgenic tobacco plants were created by Maria Ageyeva and Anna Brilkina together with the Timiryazev Institute of Plant Physiology Russian Academy of Sciences.

Dry seeds of tobacco were placed in an individual pot with a size of 7 × 7 × 7 cm and were grown for 5 weeks. Potato plants were obtained by cuttings from donor plants, which were grown from tubers. Potato shoots with an apical bud and 4 lateral buds were cut, then rooted in water for 1 week, grown in pots (9 × 9 × 10 cm) with soil for 2 weeks, after which the plants were used in the experiments. The volume of soil in pots was the same for tobacco and potato. The potato and tobacco plants were grown in soil containing 220 mg/L NH_4_^+^NO_3_, 200 mg/L P_2_O_5_, 250 mg/L K_2_O, with a pH in the range of 5.5–6.5. Watering was carried out once every 3 days by tap water. The plants were cultivated in an environmentally controlled room under 16/8 h (light/dark) photoperiod at 24 °C with a cool-white and pink luminescent tubular lamp (FLUORA OSRAM, Munich, Germany; 60 μmol/m^2^ s).

Salt stress was caused by treatment of 400 mM NaCl (volume of solution was 100 mL). Concentration of NaCl solution was selected earlier as it caused a change in photosynthesis. The treatment by solution without NaCl (tap water) was used as the control. Solutions poured into the pot closed at the bottom. Concentration of sodium in soil is shown in Appendix A.

### 4.2. Chlorophyll Fluorescence Measurement

The parameters of the light-driven reactions of photosynthesis were recorded using PlantExplorerPro+ (PhenoVation, Wageningen, The Netherlands). The parameters of Photosystem II fluorescence in the dark-adapted state (F_v_/F_m_), the quantum yield of photosystem II photochemical reactions (Ф_PSII_) and non-photochemical fluorescence quenching (NPQ) were calculated using Data Analysis Software Version 5.6.7-64b of the device according to the following equations [56]:F_v_/F_m_ = (F_m_ − F_0_)/F_m_,(1)
Ф_PSII_ = (F_m_′ − F)/F_m_,(2)
NPQ = (F_m_ − F_m_′)/F_m_′,(3)
where F_0_ is the image of the minimum fluorescence of chlorophyll in a dark-adapted state, F_m_ is the maximum fluorescence yield, F_m_′ is the maximum fluorescence yield in light condition, F is the current fluorescence level.

The chlorophyll index (CHL-Ind) was calculated using the equation [103]:CHL-Ind = R_700_^−1^ − R_NIR_^−1^,(4)
where R_700_^−1^ is reciprocal reflectance (R_λ_)^–1^ in the spectral range λ from 520 to 550 nm and 695 to 705 nm; R_NIR_^−1^ is near infra-red reciprocal reflectance.

White actinic light was used to maintain photosynthesis and light is the sum of the photon flux from diodes with radiation maxima at 455, 525 and 660 nm. The photon flux density of actinic light was 136 µmol/m^2^ s for tobacco plants and 117 µmol/m^2^ s for potato plants. Saturating flashes were created by illumination at a wavelength of 635 nm with a photon flux density of 2881 µmol/m^2^ s. The duration of saturating flashes was 300 ms. The duration of the dark adaptation preceding the measurements was 15 min.

### 4.3. Fluorescent Imaging of Cytosolic pH

Fluorescent imaging of the cytosolic pH in tobacco and potato leaves was carried out by detecting changes in the fluorescence of the Pt-GFP sensor. Fluorescence of Pt-GFP were exited at 445–475 nm and 390–420 nm in PlantExplorerPro+ (PhenoVation, Wageningen, The Netherlands) by luminodiodes; the emitted fluorescence was detected using a CMOS sensor with a 530 ± 20 nm filter.

The dependence of Pt-GFP fluorescence on the pH in the leaves was analyzed to determine the cytosolic pH level in tobacco and potato. For determination of this dependence, potato and tobacco leaves were incubated in buffer solutions with different pH levels from 4 to 9 (0.5 increments) and 125 µM of the ionophore carbonyl cyanide 3-chlorophenylhydrazone (CCCP). The composition of the buffer solutions is described in [102]. The incubation time was 4 h for potato leaves and 6 h for tobacco leaves. After incubation images of leaf fluorescence were obtained, the resulting images were processed using Data Analysis Software Version 5.6.7-64b and FiJi [104] to calculate of the dependence of Pt-GFP fluorescence on cytosolic pH.

### 4.4. CWSI and RWC Measurement

Monitoring of changes in the crop water stress index (CWSI) of tobacco and potato leaves was performed by detecting the temperature of their surface (T), moisture (T_moisture_) and dry (T_dry_) standard using a thermal imager Testo 885 (Testo, Lenzkirch, Germany). The resulting images were processed using IRSoft software (version 4.8), and the stomatal conductance value of the leaf was obtained from the equation [105]:CWSI = (T_dry_ − T)/(T_dry_ − T_moisture_).(5)

The relative water content (RWC) in the leaves was estimated by the ratio of fresh weight (FW) and dry weight (DW) of leaves. The leaves were dried for 48 h at 70 °C to obtain the dry weight value. The water content was calculated by the equation:RWC = (FW − DW)/FW.(6)

### 4.5. Measurement of Sodium, Potassium and Chloride Content

The sodium content in potato and tobacco leaves was determined using an Ag^+^/AgCl ion-selective glass electrode ELIS-112Na (NPO Measuring Technology MT, Moscow, Russia) and an Ag^+^/AgCl reference electrode ELIS-1M3.1 filled with a solution of 3M KCl (NPO Measuring Technology MT, Moscow, Russia). The potassium content was determined using an ion-selective film electrode ELIS-121K (NPO Measuring Technology MT, Moscow, Russia) and Ag^+^/AgCl reference electrode EVL-1M3.1 filled 1M Li_2_SO_4_ solution (NPO Measuring Technology MT, Moscow, Russia). The chloride content was determined using an ion-selective film electrode ELIS-131Cl (NPO Measuring Technology MT, Moscow, Russia) and Ag^+^/AgCl reference electrode EVL-1M3.1 filled 1M KNO_3_ solution (NPO Measuring Technology MT, Moscow, Russia). The electrodes were connected to the high-impedance amplifier IPL-112 (Semiko, Novosibirsk, Russia). A standard weight of dried leaves was dissolved in 25 mL of distilled water and heated at 90 °C for 3 h. Then, the pH was adjusted with ammonia vapors above 8 pH units and the concentration of sodium, potassium and chloride ions were measured by electrodes. The ion content was expressed in mmol/g dry weight (DW).

### 4.6. Statistical Analysis

Statistical analysis was performed using GraphPad Prism 6 software. Data are represented as mean ± standard error of mean (SEM). The number of biological replicates was from 5 to 9. The data were analyzed using two-way ANOVA and Bonferroni correction. * is *p* < 0.05.

## Figures and Tables

**Figure 1 ijms-24-00491-f001:**
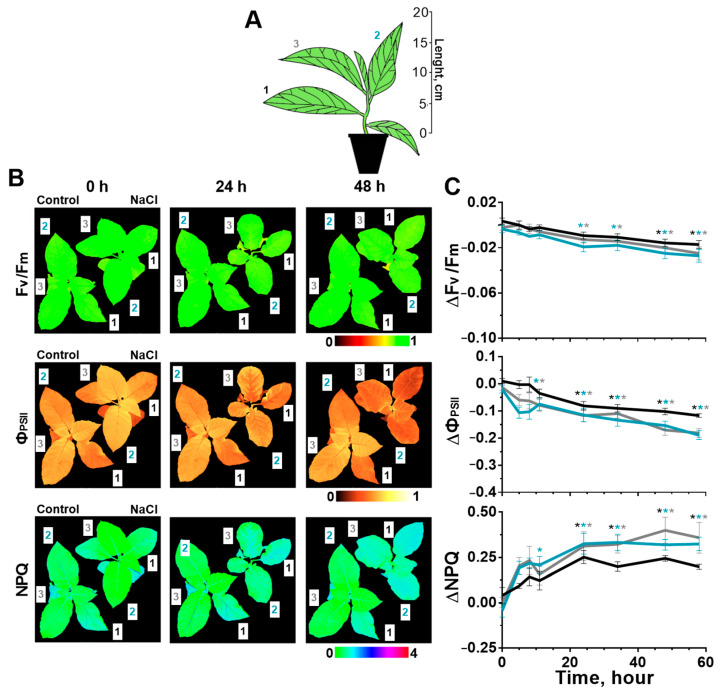
Effect of salt stress on photosynthesis in three tobacco leaves located at different height on shoot (1, 2, 3) (**A**). Changes of F_v_/F_m_, Ф_PSII_ and NPQ are demonstrated as whole-plant imaging in the pseudo-color scale in control (Control) and salt–treated (NaCl) plants at three time points (0, 24 and 48 h) (**B**), as well as presented in the form of changes relative to control values (1—black, 2—blue, 3—gray) (**C**). The data are presented as mean ± SEM (n = 9), * *p* < 0.05 control versus salt treatment.

**Figure 2 ijms-24-00491-f002:**
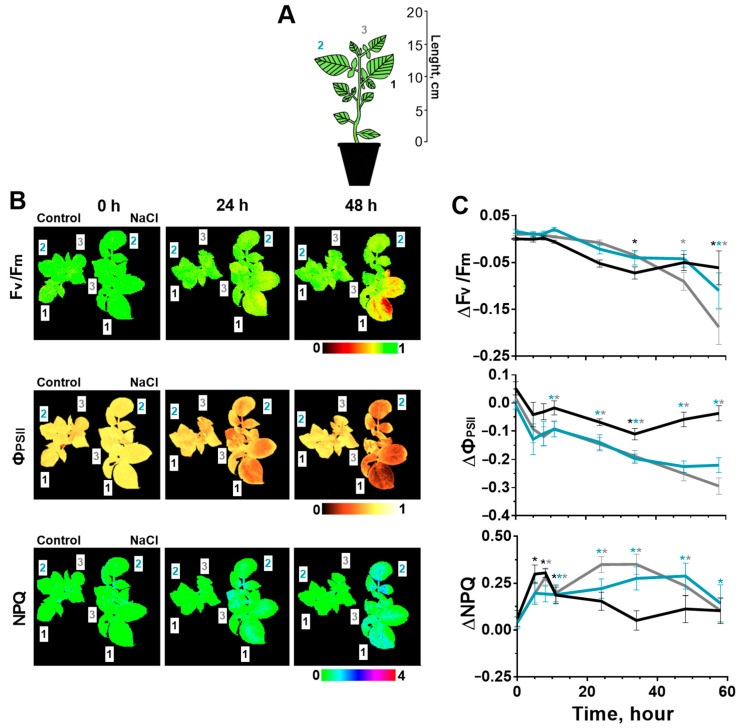
Effect of salt stress on photosynthesis in three potato leaves located at different height on shoot (1, 2, 3) (**A**). Changes of F_v_/F_m_, Ф_PSII_ and NPQ are demonstrated as whole-plant imaging in the pseudo-color scale in control (Control) and salt–treated (NaCl) plants at three time points (0, 24 and 48 h) (**B**), as well as presented in the form of changes relative to control values (1—black, 2—blue, 3—gray) (**C**). The data are presented as mean ± SEM (n = 9), * *p* < 0.05 control versus salt treatment.

**Figure 3 ijms-24-00491-f003:**
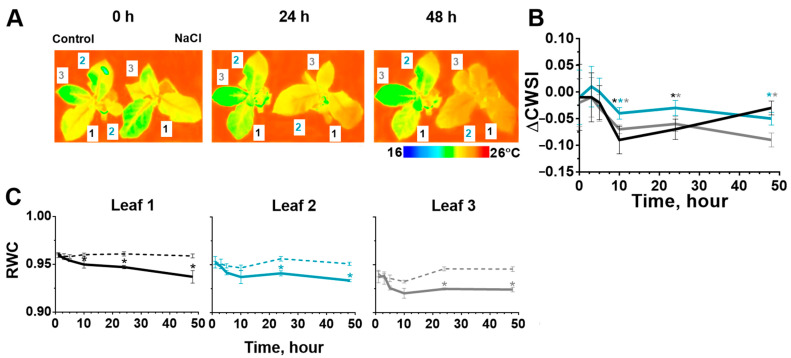
Temperature of tobacco leaf (1, 2, 3) surface (at three time points: 0, 24 and 48 h) (**A**), CWSI relative control plant (leaves: 1—black, 2—blue, 3—gray) (**B**) and RWC (control—dotted, salinity—solid) (**C**) changed during salt stress. The data are presented as mean ± SEM (n = 6), * *p* < 0.05 control versus salt treatment.

**Figure 4 ijms-24-00491-f004:**
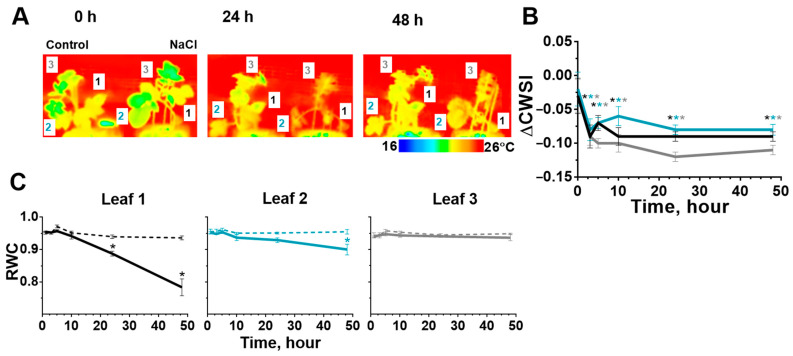
Temperature of potato leaf (1, 2, 3) surface (in at three time points: 0, 24 and 48 h) (**A**), CWSI relative control plant (leaves: 1—black, 2—blue, 3—gray) (**B**) and RWC (control—dotted, salinity—solid) (**C**) changed during salt stress. The data are presented as mean ± SEM (n = 6), * *p* < 0.05 control versus salt treatment.

**Figure 5 ijms-24-00491-f005:**
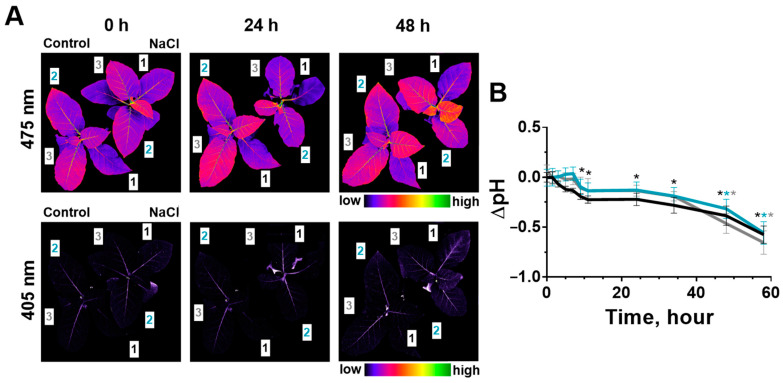
Change in fluorescence of the Pt-GFP sensor and cytosolic pH in different stratum of tobacco leaves (1, 2, 3) during salinity. (**A**) Changes in transgenic plant fluorescence (excitation of 475 and 405 nm) are demonstrated as whole-plant imaging in the pseudo-color scale in control (Control) and salt-treated (NaCl) plants at three time points (0, 24 and 48 h) of salt stress. Cytosolic pH (**B**) increased relative to control (water-treated plants) values (leaves: 1—black, 2—blue, 3—gray). The data are presented as mean ± SEM (n = 9), * *p* < 0.05 control versus salt treatment.

**Figure 6 ijms-24-00491-f006:**
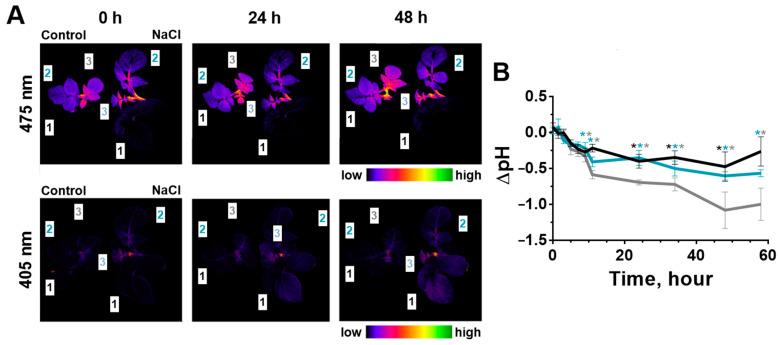
Change in fluorescence of the Pt-GFP sensor and cytosolic pH in different stratum of potato leaves (1, 2, 3) during salinity. (**A**) Changes in transgenic plant fluorescence (excitation of 475 and 405 nm) are demonstrated as whole-plant imaging in the pseudo-color scale in control (Control) and salt-treated (NaCl) plants at three time points (0, 24 and 48 h) of salt stress. Cytosolic pH (**B**) increased relative to control (water-treated plants) values (leaves: 1—black, 2—blue, 3—gray). The data are presented as mean ± SEM (n = 9), * *p* < 0.05 control versus salt treatment.

**Figure 7 ijms-24-00491-f007:**
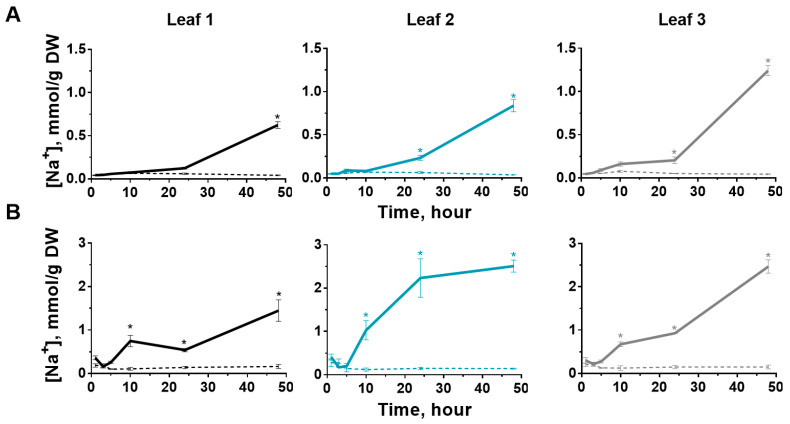
Changes in the sodium content in leaves of tobacco (**A**) and potatoes (**B**) (control—dotted, salinity—solid). The data are presented as mean ± SEM (n = 6), * *p* < 0.05 control versus salt treatment.

**Figure 8 ijms-24-00491-f008:**
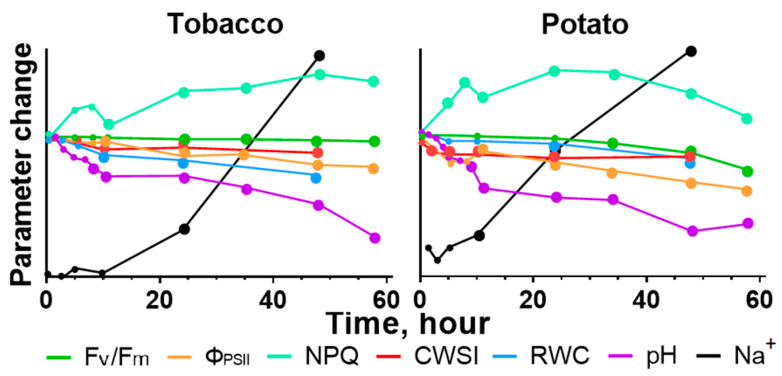
Common temporal dynamics of early (among leaves of different stratum) changes for all measured parameters for tobacco and potato plants. Small dots indicate experimental points, large dots indicate transition to significant changes of parameter relative to control condition.

**Figure 9 ijms-24-00491-f009:**
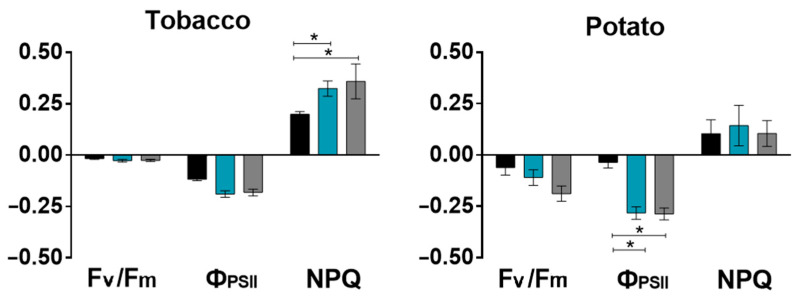
Salt-induced changes of Fv/Fm, Ф_PSII_ and NPQ depending on the leaf stratum (1—black, 2—blue, 3—gray) at 56 h after the start of salt treatment. The data are presented as mean ± SEM (n = 9), * *p* < 0.05 between the two parameters.

## Data Availability

Data are contained within the article and Appendix A.

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
