# Peer review of "Salt-Induced Changes in Cytosolic pH and Photosynthesis in Tobacco and Potato Leaves"

_ijms, 2022, doi:10.3390/ijms24010491_

Round 1
Reviewer 1 Report
Figure 4. provides a different viewing perspective,why?
Figure 8. The abscissa is the time. The ordinate is what?
Author Response
Dear Reviewer,
We are grateful to you for your comments and suggestions for improving our work. We have revised the manuscript according to your comments. Below are the answers to your questions mentioned in the review (the reviewer's comments are highlighted in bold, our answers follow in plain text).
Point 1: Figure 4. provides a different viewing perspective,why?
Answer 1: Thank you for your comment. Compared to tobacco, for potato plants fixation of leaves for thermometry was more difficult, since many plants were analyzed at the same time. We used wooden sticks and foam blocks for stem and leaves fixation which is important for leaves losing turgor. This fact explains the apparent distortion of the measurement perspective, however, the measure perspective was similar to tobacco. During measurement thermal imager Testo 885 safes temperature image (thermograms) and real photo images (Figure 1).
Point 2: Figure 8. The abscissa is the time. The ordinate is what?
Answer 2: Thank you for your comment. It was difficult for us to name the ordinate axis because it described changes of many parameters. We thought about it and, taking into account your comment, decided that it was worth adding the name "Parameter change". The updated Figure 8 was add in the new version of Manuscript.

Reviewer 2 Report
The work entitled "Salt-induced changes in cytosolic pH and photosynthesis in tobacco and potato leaves" could be interesting but needs further investigation.
1) The authors consider only one treatment concentration (400 mMNaCl) I think it would have been better to consider at least two, or at least an explanation should be given for the choice.
2) The authors must better describe the sowing conditions (which solution, what temperature and light conditions, etc.) and the development of tobacco and potato seedlings until they were transferred to the soil.
3) The soil is too generic. Please authors explain what is the composition of the soil before the addition of NH4+ NO3, 200 mg/l P2O5, 250 mg/l K2O.
4) Chlorine is a micronutrient and is as toxic as or perhaps more toxic than sodium to the plant. The work to be improved should consider the dynamics of chlorine entry in the plant and above all the concentration at which it can be inside the plant after 48 h. It is not enough to provide the bibliographic reference to state “since the rate of input of sodium and chlorine from the roots is the same [53, 96]”.
5) Figures 1B and 2B are not introduced in the results.
6) Figures 8 and 9 should be introduced into the results then discussed in the discussion.
7) Species must be written in italics.
8) The molarity through the text it is expressed as mol or M. Please change uniformly to M.
9) Why is potassium concentration assessed but calcium concentration is not? The authors base some of their conclusions on the wave of calcium (Ca) “the propagation of a remote stress signal from the root to the shoot mutually interplayed waves of calcium from the root to the shoot”.
10) In figure 1 b and 2b the time of observation reaches 48h, while in 1c e2c 60h. In addition, at 60h there are significant differences compared to 48h regarding the parameters considered. Please give an explanation and discuss the results.
11) In line 400 why different incubation times?
12) “This work provides the basis for studying the propagation of distant signal from root to shoot and cytosol pH changes in the plant in the first hours and days after the start of salinity and their effect on photosynthesis”. This last sentence must be changed because the work does not analyze the propagation of the signal but only provides assumptions.
In the abstract remove the abbreviations in lines 19, 20, 21, 22
Introduce abbreviations the first time they appear then use the abbreviation
Line 32: change [2], [3] to [2, 3]
Line 45: change NSCC in (NSCC)
Line 97: change to The analysis of photosynthesis parameters: non-photochemical quenching (NPQ), quantum yield of photosystem II (ФPSII ), Photosystem II fluorescence in the dark-adapted state (Fv/Fm) and chlorophyll index (CHL-Ind)
Line 103: photosystem II (PSII) activity
Lines 14-15, 30-32, 98-100, 231-33: unclear sentences
Lines 233-234: remove (Figure 1C) and (Figure 2C)
Line 361: rooted in which water? Distilled?
Line 378: Fo or F0?
Line 398: which buffer solution? to specify
Line 399: CCCP?
Figure S4: change sodium with potassium
Author Response
Response to Reviewer 2
Dear Reviewer,
We are grateful to you for your comments and suggestions for improving our work. We have revised the manuscript according to your comments. Below are the answers to your questions mentioned in the review (the reviewer's comments are highlighted in bold, our answers follow in plain text).
Point 1) The authors consider only one treatment concentration (400 mMNaCl) I think it would have been better to consider at least two, or at least an explanation should be given for the choice.
Answer 1. Initially, test experiments were conducted to assess the added salt concentration. We tested several concentrations (75, 100, 200, 300 and 400 mM) of the added NaCl solution to find out which of them leads to changes in photosynthesis parameters. The change in photosynthesis was evaluated after 12 and 24 hours. Only the last concentration had a noticeable/pronounced effect on photosynthesis, so we chose it as the working one in our experiments. In most previous studies experiments with salt stress are carried out on plants living in hydroponic conditions or in vitro. In this case, the exact final salt concentration is known and it is convenient for the researcher to control the salinity process (accurately increase the salt concentration, remove salt from the solution, etc.). In experiments using hydroponics for tobacco, different concentrations of NaCl are usually used: 100 mM (https://doi.org/10.1104%2Fpp.81.2.454), 200 mM (http://dx.doi.org/10.1093/jxb/ers135; https://doi.org/10.3389/fpls.2016.00217), and 250 (https://doi.org/10.1371/journal.pone.0159588). For potato plants – 300 mM (10.1016/j.plantsci.2005.03.021), 100 mM (https://doi.org/10.1007/s11120-020-00708-z), 60 mM (https://doi.org/10.3389/fpls.2018.00737). In vitro experiments researchers usually use concentration of NaCl in culture medium for tobacco: 400 mM (https:// doi.org/10.3390/plants10061102 ; https://doi.org/10.1073/pnas.2034667100), 200 mM (doi:10.1371/journal.pone.0076392; http://dx.doi.org/10.1007/s11103-012-9928-8), 300 mM (https://doi.org/10.3389/fpls.2016.00217); for potato - 120 mM (https://doi.org/10.1007/BF02854214), 250 mM (http://dx.doi.org/10.1590/1413-7054202044004220; https://www.jstor.org/stable/26525339), 300 mM (https://doi.org/10.1186/s40538-022-00286-3). These conditions are not natural for the cultivation of most plants. In the conditions of hydroponics, an additional effect of hypoxia of the roots is often created which does not allow for the correct interpretation of the results. Normally, the soil contains NaCl no more than 4 dS/m soil (approximately 40 mM), since from this concentration the effects of salinity begin to manifest (https://doi.org/10.1016/B978-0-12-818095-2.00007-2; https://doi.org/10.1016/B978-0-12-814719-1.00014-8). In our work, we studied plants growing on the soil in pots. Salinity was caused by a single watering of the soil 400 mM NaCl solution per pot. Part of solution poured into the pot closed at the bottom. The sodium concentration in the soil was low (Figure 1) but it was enough to cause the physiological effects observed in our study. Sodium concentration in soil was 0,16 mmol/g or approximately 160 mM at 1 hour of salt treatment.
We added information to the Methods in 4.1 Plant Material and Salt Treatment, as well as in Supplementary – Figure S6.
Point 2) The authors must better describe the sowing conditions (which solution, what temperature and light conditions, etc.) and the development of tobacco and potato seedlings until they were transferred to the soil.
Answer 2: Tobacco plants were obtained from seeds. Tobacco seeds were sown one by one in individual pots. In the experiments, 5-week-old tobacco plants were used with three fully formed leaves. Length of plant was 15-20 cm.
Potato plants were obtained by cuttings from donor plants grown from tubers. Potato shoots with an apical bud and 4 lateral buds were cut off, placed in water (prepared tap water) for 1 week. During this time, the cuttings formed roots (two and more) and these cuttings were planted in pots with soil. After 2 weeks, potato plants reached a height of 15-20 cm (like tobacco) and had three tiers of leaves. These plants were used in experiments.
Watering of plants was carried out 1 time in three days with tap water. The plants were cultivated in an environmentally controlled room under 16/8 h (light/dark) photoperiod at 24 °C with a cool-white and pink fluorescence tubular lamp (FLUORA OSRAM, Munich, Germany; 60 μmol/m2s).
We added changes to the Methods, 4.1 Plant Material and Salt Treatment.
Point 3) The soil is too generic. Please authors explain what is the composition of the soil before the addition of NH4+ NO3, 200 mg/l P2O5, 250 mg/l K2O.
Answer 3. “220 mg/l NH4+NO3, 200 mg/l P2O5, 250 mg/l K2O” is the content of elements in the soil specified by the manufacturer Veltorf, Velikiye Luki, Russia. This is a standard soil for our research, for example, we used it in research when growing potato plants (https://doi.org/10.3390/agriculture11111131). We measured the sodium content in it, it was 0.003 mol/g DW (~1 mM) (Fig. 1).
Point 4) Chlorine is a micronutrient and is as toxic as or perhaps more toxic than sodium to the plant. The work to be improved should consider the dynamics of chlorine entry in the plant and above all the concentration at which it can be inside the plant after 48 h. It is not enough to provide the bibliographic reference to state “since the rate of input of sodium and chlorine from the roots is the same [53, 96]”.
Answer 4. Thank you for your comment, we conducted additional experiments: we measured the concentration of sodium and chloride ions simultaneously at 24 and 48 hours after salt treatment (Figure 2). The concentration of chlorine during salinity increased which was similar dynamics of sodium accumulation. We agree that chlorine can also have a toxic effect.
Normally, the chlorine concentration in the shoots is 1-20 mg/g DW (https://doi.org/10.1007/s00425-003-1137-x; https://doi.org/10.1016/j.plantsci.2018.02.014). Tobacco (Nicotiana tabacum) can accumulate up to 50 mg/g DW during 5 mM Cl treatment (doi:10.1093/pcp/pcy071). Also, accumulation of Cl- in tobacco leaves up to 33 mg/g DW is shown during 800 mg Cl- / kg soil treatment (https://doi.org / 10.3390/agronomy11040736).
These results were added to the Manuscript and Supplementary as Figure S4.
Point 5) Figures 1B and 2B are not introduced in the results.
Answer 5. We have added a description of these figures in line 101.
Point 6) Figures 8 and 9 should be introduced into the results then discussed in the discussion.
Answer 6. We have placed Figures 8 and 9 in the Discussion after the first mention of these figures. Figures 8 and 9 show the data presented in figures in Results section. Figures 8 and 9 content summary results, and they were additionally grouped and calculated. We believe that location of these figures in Discussion contributes to the reading of the manuscript.
Point 7) Species must be written in italics.
Answer 7. Thank you for your comment, we fixed this error.
Point 8) The molarity through the text it is expressed as mol or M. Please change uniformly to M.
Answer 8. Thank you for your comment. We corrected this error where it was required. In the case amount of the substance we did not fix mol.
Also we recalculated ion concentrations and corrected results. We updated Figure 7 and S5. Thank you for drawing our attention to M and mol which allowed us to fix serious error in our calculations and to improve our manuscript.
Point 9) Why is potassium concentration assessed but calcium concentration is not? The authors base some of their conclusions on the wave of calcium (Ca) “the propagation of a remote stress signal from the root to the shoot mutually interplayed waves of calcium from the root to the shoot”.
Answer 9. The increased sodium content in the soil causes leakage of potassium from the cell. The cell contains many enzymes controlled by K+ and involved in primary metabolism such as the Calvin cycle, the phenylpropanoid pathway, glycolysis and synthesis of polyamine and starch (https://doi.org/10.1071/FP21153 ; https://doi.org/10.1016/j.xinn.2020.100017 ). Sodium and potassium are often measured together, for example, in articles (doi:10.1371/journal.pone.0060183; doi:10.1093/jxb/err420; https://doi.org/10.1071/fp05080; doi: 10.3389/fpls.2014.00787; http://dx.doi.org/doi:10.1016/j.envexpbot.2017.07.003).
Calcium is interesting for us as signal molecule. The calcium content in the cytosol is normally low, and when the signaling pathway is activated, it is released from intracellular or extracellular depots. It is important to study the dynamics of intracellular calcium. Change of calcium dynamic occurs under the effect of many stress factors. However, we measured the total ion content in the plant, and we could not conclude from the change in the total calcium content about the formation of a calcium signal in the cell. Due to the lack of calcium measurement results, we have shortened this theme in Discussion.
Point 10) In figure 1 b and 2b the time of observation reaches 48h, while in 1c e2c 60h. In addition, at 60h there are significant differences compared to 48h regarding the parameters considered. Please give an explanation and discuss the results.
Answer 10. Figures 1B and 2B show images of plant leaves obtained by the method of PAM-fluorimetry in PlantExplorerPro+ (PhenoVation, the Netherlands), and they illustrate photosynthetic activity of photosystem II. Images are presented in a pseudo-color scale and graphically demonstrate changes in photosynthetic activity. In long-term experiments (more than 1 day) it should be mention that any physiological processes in plants including photosynthesis are subject to circadian rhythms. Therefore, measurements at exactly the same time of day are important to take. Figures 1B and 2B show images illustrating changes in the parameters Fv/Fm, ФPSII and NPQ after 24 and 48 hours of experiment in control and experimental plants. In these images it can be seen that the parameters change more strongly in salt-treated plants. We demonstrate dynamics of the parameters changes relative control values because some drift of these parameters are observed. And we show only three images in Figure 1B and 2B illustrating 3 time points on timeline because a greater number of images can be difficult to understand. In Figures 1C and 2C we marked all experimental points, and the changes are already noticeable at the time point of 48 hours.
Point 11) In line 400 why different incubation times?
Answer 11. We selected time of incubation in buffer solutions for each species – potato and tobacco. Selection of incubation time was from 1 hour to 24 hours. The incubation time was selected according to two parameters: 1. The solution does not violate the structure and turgor of the leaf (assessed visually); 2. Incubation does not affect the uniformity of the fluorescent signal of Pt-GFP, and there is no formation of too bright or dim zones of fluorescence in leaf. Based on these parameters, an incubation time of 4 hours was chosen for potato leaves, and 6 hours - for tobacco. The differences can be due to the specifics of the leaves of plants of each species.
Point 12) “This work provides the basis for studying the propagation of distant signal from root to shoot and cytosol pH changes in the plant in the first hours and days after the start of salinity and their effect on photosynthesis”. This last sentence must be changed because the work does not analyze the propagation of the signal but only provides assumptions.
Answer 12. Thank you for your comment, we have shortened this section in the Discussion.
Point: In the abstract remove the abbreviations in lines 19, 20, 21, 22. Introduce abbreviations the first time they appear then use the abbreviation
Answer: It is generally accepted that the Abstract and the main text of the manuscript are independent parts of the article. They imply independent reading. Therefore, abbreviations are entered twice.
Point: Line 32: change [2], [3] to [2, 3]
Answer: Thank you for your comment, we have changed this.
Point: Line 45: change NSCC in (NSCC)
Ответ: Thank you for your comment, we have changed this.
Point: Line 97: change to The analysis of photosynthesis parameters: non-photochemical quenching (NPQ), quantum yield of photosystem II (ФPSII ), Photosystem II fluorescence in the dark-adapted state (Fv/Fm) and chlorophyll index (CHL-Ind)
Answer: We took into account your comment and made sentence with abbreviations in the text at the first mention of them. Thank you for your comment!
Point: Line 103: photosystem II (PSII) activity
Answer: We have previously introduced an abbreviation for photosystem 2: line 67.
Point: Lines 14-15, 30-32, 98-100, 231-33: unclear sentences
Answer: Thank you for your comments, we corrected these sentences
Point: Lines 233-234: remove (Figure 1C) and (Figure 2C)
Answer: We believe that links to these images are necessary, as they help the reader to navigate exactly in results which are being discussed. We suggest leaving them in the text.
Point: Line 361: rooted in which water? Distilled?
Answer: Potato cuttings were rooted in tap water. Added an addition to the Methods, Plant Material and Salt Treatment.
Point: Line 378: Fo or F0?
Answer: Thank you for your comment, we fixed the uncorrected symbol.
Point: Line 398: which buffer solution? to specify
Answer: Line 401 contains a link where the composition of buffer solutions is described. We decided not to duplicate this information if it was already described earlier. And we can give this information here.
Leaves were incubated for 4 h (potato) or 6 h (tobacco) in buffer solutions with different pH levels (4.0, 4.5, 5.0, 5.5, 6.0, 6.5, 7.0, 7.5, 8.0, 8.5, 9.0) to obtain the dependence of Pt-GFP fluorescence on pH. Buffer solutions contained 125 μM the protonophore carbonyl cyanide 3-chlorophenylhydrazone (CCCP). Buffer solutions had the following composition: pH 4.0, 4.5, 5.0 and 5.5 (40 mM sodium citrate, 40 mM MES); 6.0 (40 mM sodium citrate, 40 mM MES, 40 mM MOPS); 6.5 and 7.0 (40 mM sodium citrate, 40 mM MES, 40 mM MOPS, 40 mM TRIS); 7.5 (40 mM MOPS, 40 mM TRIS); and 8.0, 8.5 and 9.0 (80 mM TRIS). The base of all buffers was a standard solution (1 mM NaCl, 0.1 mM KCl, 0.1 mM CaCl2). The solutions were buffered with 1M NaOH or 1M HCl.
Point: Line 399: CCCP?
Answer: Thank you for the comment, we added the full name of the substance.
Point: Figure S4: change sodium with potassium
Answer: Thank you for your comment we fixed this error.

Round 2
Reviewer 1 Report
Accept after minor revision.
Author Response
Dear Reviewer,
We are grateful to you for checking and evaluating our manuscript. Your comments helped make our manuscript better.
Reviewer 2 Report
In the abstract NPQ, RWC, CWSI are reported only once so it is useless to abbreviate them!
The sowing conditions (culture medium or water in which the seeds were sown), are still missing. Were they kept in the dark? At what temperature? I would recommend adding them
Author Response
Dear Reviewer,
We are grateful to you for your comments and suggestions for improving our work. We have revised the manuscript according to your comments. Below are the answers to your questions mentioned in the review (the reviewer's comments are highlighted in bold, our answers follow in plain text).
Point 1: In the abstract NPQ, RWC, CWSI are reported only once so it is useless to abbreviate them!
Answer 1: We deleted these abbreviations. We are grateful to you for your comment.
Point 2: The sowing conditions (culture medium or water in which the seeds were sown), are still missing. Were they kept in the dark? At what temperature? I would recommend adding them
Answer 2: The procedure for obtaining the tobacco plant was as described below. Dry seeds of tobacco were placed in an individual pots with a size of 7x7x7 сm and were grown for 5 weeks to plants which was used in experiments. Thus, experimental tobacco plants were not transplanted during the growth process from seeds. Conditions of seed germination were same conditions of plant cultivation: 16/8 h (light/dark) photoperiod at 24 °C with a cool-white and pink luminescent tubular lamp (FLUORA OSRAM, Munich, Germany; 60 μmol/m2s). We added this information in the Methods, lines 364-365.
We are grateful to you for checking and evaluating our manuscript. Your comments helped make our manuscript better.
